# Cataract Surgery in Elderly Subjects with Heterozygous Familial Hypercholesterolemia in Prolonged Treatment with Statins

**DOI:** 10.3390/jcm10163494

**Published:** 2021-08-08

**Authors:** Victoria Marco-Benedí, Martín Laclaustra, Rosa M. Sánchez-Hernández, Emilio Ortega-Martínez de Victoria, Juan Pedro-Botet, Jose Puzo, Fernando Civeira

**Affiliations:** 1Lipid Unit, Hospital Universitario Miguel Servet, IIS Aragón, CIBERCV, 50009 Zaragoza, Spain; vmarcobenedi@gmail.com; 2Departamento de Medicina, Psiquiatría y Dermatología, Universidad de Zaragoza, 50009 Zaragoza, Spain; 3Endocrinology Department, Hospital Insular Universitario de Gran Canaria, 35016 Las Palmas de Gran Canaria, Spain; rosamariasanher@gmail.com; 4Lipid Clinic, Endocrinology Department, Hospital Clinic, CIBEROBN, 08036 Barcelona, Spain; EORTEGA1@clinic.cat; 5Lipid Unit, Hospital del Mar, Universitat Autònoma de Barcelona, 08003 Barcelona, Spain; JPedrobotet@parcdesalutmar.cat; 6Lipid Unit, Hospital San Jorge, 22004 Huesca, Spain; josepuzo@gmail.com

**Keywords:** cataracts, heterozygous familial hypercholesterolemia, statins, lipid-lowering treatment

## Abstract

Background: Cataracts are the main cause of blindness and represent one fifth of visual problems worldwide. It is still unknown whether prolonged statin treatment favors the development of cataracts. We aimed to ascertain the prevalence of cataract surgery in elderly subjects with genetically diagnosed heterozygous familial hypercholesterolemia (HeFH) receiving statin treatment for ≥5 years, and compare this with controls. Methods: This is an observational, multicenter, case–control study from five lipid clinics in Spain. We collected data with the following inclusion criteria: age ≥65 years, LDL cholesterol levels ≥220 mg/dL without lipid-lowering drugs, a pathogenic mutation in a candidate gene for HeFH (LDLR, APOB, or PCSK9) and statin treatment for ≥5 years. Controls were selected from relatives of HeFH patients without hypercholesterolemia. Linear and logistic regressions based on generalized linear models and generalized estimating equations (GEE) were used. Cataract surgery was used as a proxy for cataract development. Results: We analyzed 205 subjects, 112 HeFH, and 93 controls, with a mean age of 71.8 (6.5) and 70.0 (7.3) years, respectively. HeFH subjects presented no difference in clinical characteristics, including smoking, hypertension, and type 2 diabetes mellitus, compared with controls. The mean duration of lipid-lowering treatment in HeFH was 22.5 (8.7) years. Cataract surgery prevalence was not significantly different between cases and controls. The presence of cataracts was associated neither with LDLc nor with the length of the statin therapy. Conclusion: In the present study, HeFH was not a risk factor for cataract surgery and prolonged statin treatment did not favor it either. These findings suggest that statin treatment is not related with cataracts.

## 1. Introduction

Cataracts cause one third of blindness worldwide, sharing the leading position with uncorrected refractive errors and glaucoma [1], and 20–25 million cataract surgical interventions are performed worldwide each year [2]. Cataracts are defined as a degradation of the optical quality of the eye due to the clouding of the crystalline lens. Several properties of the lens gradually decline with age, and accordingly, old age is the most important risk factor in cataract formation. Other common risk factors include diabetes mellitus (DM); long-term use of topical, systemic, intravitreal, inhaled, or oral corticosteroids; prior intraocular surgery; trauma; smoking; and exposition to ultraviolet-B light [3,4].

Statins are inhibitors of the enzyme HMG CoA reductase commonly used as lipid-lowering drugs [5]. They are used to lower blood cholesterol, which has proved to be a highly effective strategy to prevent cardiovascular disease in high-risk subjects. Since the description of vision loss due to irreversible cataracts caused by triparanol, the first synthetic cholesterol-lowering drug [6], some reports have associated the use of statins with the development of cataracts, although with conflicting results. A recent meta-analysis including observational studies concluded that statins slightly increase the risk of cataracts [5,7], whereas in randomized clinical trials, statins did not increase the risk of cataracts [8]. This topic was recently reviewed by several councils of the American Heart Association, and their conclusion was that statins in clinical use do not increase the risk of cataracts [9]. However, these observational studies and clinical trials were performed in populations in which the prevalence of cataracts is not very high, since they excluded patients ≥75 years of age, and in which the use of statins is limited to only a few years, usually less than 5 years.

Familial hypercholesterolemia (FH) is a monogenic disease characterized by an abnormal increase in low-density lipoprotein cholesterol (LDLc) levels from birth and subsequent high-risk of coronary disease. Only heterozygous FH (HeFH) patients reach older age. For this reason, HeFH is a group of patients in whom a high dose of potent statins has been used for decades. Thus, elderly HeFH subjects, having undergone decades of potent lipid-lowering therapy, may be an attractive population model to explore unexpected side effects [10,11]. The development of cataracts in HeFH has not been studied. We aimed to study the association between cataracts and statin use in a group of elderly HeFH under prolonged statin treatment, and compare this with controls.

## 2. Methods

### 2.1. Study Characteristics

This is an observational, multicenter, case–control study. We studied HeFH cases and controls from five lipid clinics in Spain. The protocol has been previously published [12] and was designed to explore non-coronary morbidities in elder HeFH. In brief, we recruited subjects with age ≥65 years; men and women, with a pathogenic mutation in a candidate gene for FH (*LDLR, APOB, or PCSK9*) in the subject or in a first-degree relative; LDLc levels ≥220 mg/dL without lipid-lowering therapy; and statin treatment for ≥5 years. Controls were selected from relatives of HeFH patients, requiring the absence of hypercholesterolemia (LDLc <190 mg/dL without lipid-lowering treatment). All subjects gave their informed consent for inclusion before they participated in the study. The study was conducted in accordance with the Declaration of Helsinki, and the protocol was approved by the Ethics Committee of Aragon (CEICA) #PI19/440.

### 2.2. Assessments

In a clinical interview, we collected age, gender, ethnicity, smoking habit, and personal history of hypertension, diabetes mellitus, cataract surgery and cardiovascular heart disease. Cataract surgery was confirmed through reviewing the medical records. In addition, information of lipid-lowering treatment, such as statins, ezetimibe, and subtilisin-convertase proprotein/kexin type 9 inhibitors (PCSK9i), was collected. The recorded data included the type of drug and dose prescribed in the current treatment, the most common treatment used during the subject’s lifetime, dose, and time when lipid-lowering treatment started. Body mass index (BMI) was calculated as weight in kilograms divided by the square of height in meters.

### 2.3. Statistical Analyses

We analyzed the association of cataracts with HeFH using generalized estimating equations (GEE), using logistic models with several levels of adjustment: unadjusted, adjusted for sex and age, and additionally adjusted for untreated LDLc concentration. Data for cases and controls are shown as mean (standard deviation) or percentage. The study of the influence of LDLc and statin treatment was analyzed within strata (cases and controls separately) with generalized linear models. All the analyses were performed with the statistical software R version 3.4.4. and the package “gee” version 4.13.19 (R Foundation, Auckland, New Zealand).

## 3. Results

Data were collected for 205 subjects (112 HeFH and 93 controls) aged 71.8 (6.5) and 70.0 (7.3) years, respectively. All cases and controls were Caucasian. The number (percentage) of women and men was 74 (66.1%) and 38 (33.9%) in cases and 48 (51.6%) and 45 (48.4%) in controls, respectively. There were no differences in clinical characteristics between cases and controls, with the exception of data on history of cardiovascular diseases, which were more frequent in HeFH subjects’ blood-relatives (45.0% vs. 25.8%) and in themselves (27.7% vs. 16.1%); *p* < 0.05 in both cases. Similarly, the concentration of LDLc was higher in HeFH cases: 314 vs. 138 mg/dL (*p* < 0.01). The mean duration of statin treatment use in HeFH was 22.5 (8.7) years. Ninety-nine out of the 112 (88.4%) HeFH patients were on high-potency statins (atorvastatin 40–80 mg and rosuvastatin 20–40 mg). We did not observe any differences in smoking, hypertension, and DM between cases and controls [12]. A history of cataract surgery was present in 25.2 % of cases and 16.1% of controls, without difference in gender distribution. This difference was not statistically different either in the unadjusted test or after adjusting for age and sex, or additionally for LDLc (Table 1). When cases were classified according to the presence or absence of cataract surgery, there were no differences in clinical variables, except for age, which was higher in HeFH with cataract surgery (Appendix A). We also analyzed the association of age, duration of statin treatment, and LDLc without lipid-lowering treatment with cataract surgery. Age was strongly associated with a relative risk of 2.06 (CI 1.09, 4.02) per 10 years among cases and 2.57 (CI 1.13, 6.28) among controls. The duration of statin treatment (studied among cases) and LDLc without lipid-lowering treatment did not show any association with cataract surgery (Table 2).

## 4. Discussion

Our study shows that the prevalence of cataract surgery is not increased in elderly people with HeFH undergoing lipid-lowering treatment for more than 20 years. This suggests that neither severe hypercholesterolemia nor prolonged use of statins are relevant risk factors in the development of cataracts. To our knowledge, this is the first study that explores the presence of cataracts in this population, a paradigmatic subgroup of patients in whom treatment with a high dose of potent statins is the first line of treatment.

The present study has the strength of being carried out in a group of patients with three main differential characteristics with respect to previous studies: the studied subjects had very high concentrations of LDL-C from birth, they used high doses of statins for >20 years, and they are all aged ≥65 years—that is, they are a group with a high incidence of cataracts. Our study agrees that age is the main factor associated with the development of cataracts. Importantly, our data confirm the results of a previous analysis on the safety of prolonged used statins with respect to cataract development and they pose doubts on the suggested association of cholesterol concentration with cataracts. Preclinical studies showed that cholesterol has an important role in membrane integrity, and it was supposed that the inhibition of cholesterol synthesis caused cataract development [7]. In addition, it must be taken into account that statins exert their benefits across a wide spectrum of ophthalmic conditions through its hypocholesterolemic and pleiotropic effects, that may contribute to making them safe with respect to cataracts [5].

## 5. Conclusions

Cataract surgery prevalence was not significantly different between HeFH and controls. The presence of cataracts was associated neither with LDLc nor with the length of the statin therapy. In the present study, HeFH was not a risk factor for cataracts and prolonged statin treatment did not favor cataract development.

## Figures and Tables

**Table 1 jcm-10-03494-t001:** Clinical and laboratory characteristics of controls and cases.

	Controls *N* = 93	Cases *N* = 112	*p*-Value
Age (years)	70.0 (7.3)	71.8 (6.5)	0.038
Sex, women % (*n*)	51.6 [48]	66.1 [74]	0.050
Body mass index (Kg/m^2^)	29.0 (4.7)	28.3 (4.0)	0.265
Untreated LDLc (mg/dL)	138.0 (31.7)	314.2 (71.4)	<0.001
Cataract surgery % (*n*)	16.1 [15]	25.2 [28]	0.121

Continuous data expressed as mean (SD); categorical data are expressed as percentages [count]. LDLc: low-density lipoprotein cholesterol. *p*-values from linear and logistic regressions based on generalized estimating equations (GEE) with exchangeable variance structure, unadjusted.

**Table 2 jcm-10-03494-t002:** Influence of age, LDL-cholesterol, and years under statin treatment on cataract surgery in controls and in HeFH.

	Controls		Cases	
	OR (95% CI)	*p*-Value	OR (95% CI)	*p*-Value
Model 1				
Per each 10 year of age	2.49 (1.12,5.87)	0.028	2.03 (1.07,3.96)	0.031
Model 2				
Per each 10 year of age	2.57 (1.13,6.28)	0.028	2.06 (1.09,4.02)	0.028
Per each 10 mg/dL of cLDL	0.89 (0.74,1.07)	0.228	1.00 (0.94,1.07)	0.905
Model 3				
Per each 10 year of age	-	-	1.83 (0.92,3.69)	0.082
Per each 10 year of lipid-lowering use	-	-	1.01 (0.59,1.53)	0.967

Linear and logistic regressions based on generalized linear models (GLM) adjusted for sex. CI: confidence interval. *p*-values in each section obtained from a single model, adjusted for sex.

## Data Availability

Data available on request from the authors.

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
