# Peer review of "Cataract Surgery in Elderly Subjects with Heterozygous Familial Hypercholesterolemia in Prolonged Treatment with Statins"

_jcm, 2021, doi:10.3390/jcm10163494_

Round 1

Author Response

Answers:

We welcome the Reviewer's comments and have made the following changes based on his/her comments and suggestions.

  1. Ethnicity was recorded in the clinical interview. This issue has been included in the methods section.
  2. All cases and controls were Caucasian. This sentence has been included in the Results section
  3. The number and percentage of women/men distribution in cases and controls have been included in the Results section. “All cases and controls were Caucasian. The number [percentage] of women and men was 74 [66.1%] and 38 [33.9%] in cases and 48 [51.6%] and 45 [48.4%] in controls, respectively”.
  4. The findings were not different between women and men. This issue has been included in the Results section. “History of cataract surgery was present in 25.2 % of cases and 16.1% of controls without difference in gender distribution. This difference was not statistically different neither in the unadjusted test, nor after adjusting for age and sex, or additionally for LDLc (Table 1)”.

Methods Section Inclusion criteria: Age over 65 years, (Gender specification not indicated)

Answer

Both genders were included. This issue has been included in the Methods section.

Statistical Analysis Section

The analysis indicated adjustment for sex, but the table contains only female population. No male population

Answer

In table 2, both men and women are included in the analysis and the results are adjusted for sex as indicated.

Reviewer 2 Report

This paper reports and observational case-control study to examine whether familial hypercholesterolemia and statins are associated with cataracts. The results do not show an association. The paper is well written with just a few grammatical issues. There are no major concerns.

Minor concerns:

  1. Statistical Analysis should include how data are reported (e.g., mean and standard deviation)
  2. p. 2: “satins” should be “statins”
  3. p. 3 (line 55): “…due to the clouding of the crystalline.” Should be “…due to the clouding of the crystalline lens.”
  4. p.7 (line 148): “…, and old age, that is group with high…” is grammatically incorrect.

Author Response

This paper reports and observational case-control study to examine whether familial hypercholesterolemia and statins are associated with cataracts. The results do not show an association. The paper is well written with just a few grammatical issues. There are no major concerns.

 Answer

We thank the reviewer for his positive comments on our work.

Minor concerns:

  1. Statistical Analysis should include how data are reported (e.g., mean and standard deviation)

Answer

The following sentence is included in the Statistical Analysis

“Data for cases and controls are shown as mean (standard deviation) or percentage”

  1. p. 2: “satins” should be “statins”

Answer

This mistake has been corrected in the current version.

  1. p. 3 (line 55): “…due to the clouding of the crystalline.” Should be “…due to the clouding of the crystalline lens.”

Answer

This correction has been included in the current version.

  1. p.7 (line 148): “…, and old age, that is group with high…” is grammatically incorrect.

Answer

This sentence has been changed. In the new version reads as follows:

“The present study has the strength of being carried out in a group of patients with three main differential characteristics with respect to previous studies: the studied subjects had very high concentrations of LDL-C from birth, they used high-dose statins for >20 years and they are all age ≥65 years, that is, they are a group with a high incidence of cataracts”